# Carbon Peeling Laser Treatment to Improve Skin Texture, Pores and Acne Lesions: A Retrospective Study

**DOI:** 10.3390/medicina58111668

**Published:** 2022-11-18

**Authors:** Claudio Conforti, Stefania Guida, Caterina Dianzani, Piergiorgio Turco, Vito Cazzato, Iris Zalaudek, Domenico Piccolo

**Affiliations:** 1Dermatology Clinic, University of Trieste, Maggiore Hospital, Piazza Ospitale 1, 34125 Trieste, Italy; 2Dermatology Clinic, Vita-Salute San Raffaele University, 20132 Milan, Italy; 3Plastic and Reconstructive Surgery Unit, Campus Biomedico University of Rome, 00128 Rome, Italy; 4Department of Plastic and Reconstructive Surgery, Università Degli Studi di Napoli Federico II, Via Pansini 5, 80131 Napoli, Italy; 5Plastic and Reconstructive Surgery Unit, Department of Medical, Surgical and Health Sciences, University of Trieste, 34127 Trieste, Italy; 6Skin Center Avezzano, Private Practice, Avezzano, L’Aquila, Via Saragat 51, 67051 Avezzano, Italy

**Keywords:** laser, carbon peel, acne, wrinkles, rejuvenation

## Abstract

Carbon peel laser treatment has been described for the improvement of skin texture, with pore reduction and acne lesion treatment. The technique consists of applying a carbon mask to the face for about ten minutes followed by laser irradiation with a Q-switched 1064 nm laser. This mechanism of action seems to be related to small carbon molecules binding both the corneocytes and serum within the hair follicles; the effect of the laser eliminates carbon bound to skin particles and the high temperature generated reduces sebum production by sebaceous glands and inhibits Cutibacterium acnes replication. Although this method was described 20 years ago, scientific data supporting its efficacy and safety have only recently been reported in small case series. For this reason, we performed a retrospective study including patients treated from January to May 2022 in the context of a private practice. Even if this study is limited by the low number of patients and its retrospective nature, this is the first research to show that carbon peel laser, performed with a standardized technique, is an effective and safe treatment for patients with acne lesions, showing pores and wrinkles, and is able to improve the overall skin aspect.

Carbon peel laser treatment has been described for the improvement of skin texture, with pore reduction and treatment of acne lesions. The technique consists of applying a carbon mask to the face for about ten minutes followed by laser irradiation with a Q-switched 1064 nm laser. This mechanism of action seems to be related to small carbon molecules binding both the corneocytes and serum within the hair follicles; the effect of the laser eliminates carbon bound to skin particles and the high temperature generated reduces sebum production by sebaceous glands and inhibits *Cutibacterium acnes* replication. Although this method was described 20 years ago, scientific data supporting its efficacy and safety have only recently been reported in small case series [1].

We performed a retrospective study including patients treated from January to May 2022 in the context of a private practice. All patients signed an informed consent before the procedure, authorizing data analysis. 

After cleaning the skin of each patient, a carbon mask (NaturaPeel, Quanta System, Deka, Italy) was applied to the face for 10 minutes. After that, a Picosecond Q-switched laser (EI. En. Group, Florence, Italy) was employed, using 5 mm spot and 1 J/cm^2^ fluence. A total of three carbon peel laser sessions were performed, each one month apart from the other. It was decided to also evaluate, as a secondary parameter, the improvement of thin wrinkles. In fact, stimulation of the dermis induced by laser-irradiated carbon shock waves can result in improvement of wrinkles. Its efficacy was estimated through the validated Fitzpatrick Wrinkle Assessment Scale (FWAS) for the assessment of thin/deep lines on the face (from 0 *absence of wrinkles* to 6 *deep and diffuse* wrinkles) [2], the Physician Global Aesthetic Improvement Scale (GAIS) to express overall improvement (from 1 *exceptional improvement* to 5 *worsened patient*) [3], a photographic scale for the pore assessment (score system from 1 *no obvious pores* to 6 *enlarged pores*) [4] and the Investigator Global Assessment of Acne (IGA) for the classification of acne lesions (from 0 *no inflammatory lesions* to 3 *many comedones, inflammatory lesions and nodules*) [5]. These assessments were performed 1 month after the last treatment session (T1), comparing these scores to the baseline (T0), employing Student’s *t*-test (*p* < 0.05 was considered as a cut-off for significance). Main results achieved are shown in Figure 1. Additionally, safety was assessed based on major or minor adverse events, respectively, requiring medical intervention or not, after each laser session.

A total of 32 patients (13 females; 19 males) aged 26 ± 6 years old were included in the present study. Inclusion criteria were: older than 18 years old, mild or moderate acne with poor response to conventional treatments (antibiotics ± retinoids), enlarged pores of the face and fine wrinkles; exclusion criteria were patients on oral isotretinoin treatment or with an ongoing autoimmune skin disease, pregnancy, breastfeeding and absence of informed consent. Baseline characteristics of patients are reported in Table 1. Interestingly, at T1, a significant reduction of FWAS, IGA and pores scores were observed, as compared to T0. In detail, an improvement of the mid-depth wrinkles was observed in six patients, with a reduction of the FWAS score by one point (*p* < 0.01). Furthermore, 18 patients showed a reduction of one-point on the IGA scale, especially patients with mild-to-moderate acne severity (*p* < 0.01). Additionally, the photographic analysis of pore severity showed that 88% of patients had minimal pores or complete absence at T1, as compared to T0 (*p* < 0.01). Strikingly, an overall skin improvement, as assessed by the GAIS scale, was observed in 100% of patients. This finding represents a strength for this technique, which can be effective for overall skin improvement. After each laser session, only minor adverse events, including temporary erythema, was observed in 6 subjects.

Even if this study is limited by the low number of patients and its retrospective nature, this is the first research to show that carbon peel laser, performed with a standardized technique, is an effective and safe treatment for patients with acne lesions, showing pores and wrinkles, and one that is able to improve the overall skin aspect.

## Figures and Tables

**Figure 1 medicina-58-01668-f001:**
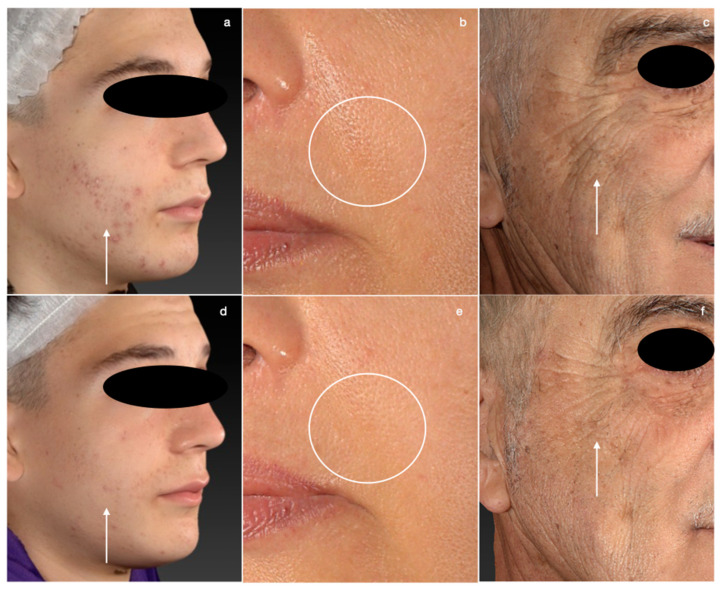
Mild acne before (**a**) and after (**d**) the carbon peel laser treatment; skin pores (**b**) of the nasolabial fold reduced after the protocol (**e**); deep wrinkles of the periocular area (**c**) and (**f**) improvement of thin and deep wrinkles of the face after the entire protocol.

**Table 1 medicina-58-01668-t001:** Characteristics of patients treated with carbon peeling laser technique.

Baseline Characteristics	Patients N = 32 *n (%)*		
Sex:			
*Females*	13 (40)		
*Males*	19 (60)		
Phototype:			
*1*	3 (9)		
*2*	6 (19)		
*3*	15 (47)		
*4*	8 (25)		
**Scores**	**Before treatment *n (%)***	**After treatment *n (%)***	** *p* ** ** ^§^ **
FWAS *:			*0.012*
*1*	4 (12)	6 (18)	
*2*	16 (50)	16 (50)	
*3*	10 (32)	10 (32)	
*4*	2 (6)	0	
Pores:			*<0.01*
*No obvious pores*	0	6 (18)	
*Minimal pores*	4 (12)	22 (70)	
*Mild pores*	18 (57)	4 (12)	
*Moderate pores*	10 (31)	0	
IGA ^#^:			*<0.01*
*Few comedons and <1 small inflammatory lesion*	0	14 (44)	
*Dozens of comedons and several inflammatory lesions*	26 (81)	16 (50)	
*Many comedons, several inflammatory lesions and <1 nodule*	6 (19)	2 (6)	
GAIS °:			
*exceptional improvement*		2 (6)	
*very improved*		14 (44)	
*improved*		16 (50)	
*unaltered*		0	
*worsened*		0	

^§^ *p*-value according to Student’s *t*-test for paired samples; * FWAS = Fitzpatrick Wrinkle Assessment Scale; ^#^ IGA = Investigator Global Assessment of Acne; ° GAIS = Global Assessment Investigator Scale.

## Data Availability

The data that support the findings of this study are available on request from the corresponding author, CC.

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
