# Peer review of "Carbon Peeling Laser Treatment to Improve Skin Texture, Pores and Acne Lesions: A Retrospective Study"

_medicina, 2022, doi:10.3390/medicina58111668_

Round 1

Reviewer 1 Report

The authors clearly presented a retrospective study on the use of carbon peel laser treatment for the improvement of skin texture, pore reduction and acne treatment. Despite such method is already widely used for such purposes, the present paper provides solid scientific background to the already published case-reports. Writing is very clear and data are well presented. Images are of good quality and provide very clear evidence of treatment efficacy.

Only some very minor changes should be done:

- line 4 "Piccolo D and MD": if the author is only MD the word AND should be replaced by a comma

- line 16: "acne lesions treatment" should be replaced by "treatment of acne lesions" or "acne-lesion treatment"

- line 35/36 the assessment of pores: "pore assessment" would probably be the preferred choice for native English speakers

- line 39 session of treatment: "treatment session" would probably be the preferred choice for native English speakers

Author Response

we thank the reviewer for his/her comments that basically helped us to improve our manuscript.

The authors clearly presented a retrospective study on the use of carbon peel laser treatment for the improvement of skin texture, pore reduction and acne treatment. Despite such method is already widely used for such purposes, the present paper provides solid scientific background to the already published case-reports. Writing is very clear and data are well presented. Images are of good quality and provide very clear evidence of treatment efficacy.

Thank you so much for your comments.

Only some very minor changes should be done:

- line 4 "Piccolo D and MD": if the author is only MD the word AND should be replaced by a comma

- line 16: "acne lesions treatment" should be replaced by "treatment of acne lesions" or "acne-lesion treatment"

- line 35/36 the assessment of pores: "pore assessment" would probably be the preferred choice for native English speakers

  • line 39 session of treatment: "treatment session" would probably be the preferred choice for native English speakers

we modified all comments as suggeste

Reviewer 2 Report

The authors should add some more information about the study group, e.g. age, the severyty of acne (mild, moderate, severe), assessment of acne using global acne grading system (GAGS.

Please, add inclusion and exclusion criteria

Why the authors have assessed Fitzpatrick Wrinkle Assessment Scale (FWAS)? What was the aim of the study? Please explain more clearly in the text. All the patients had acne problemes or you treated patients with either acne or wrinkles?

Author Response

Dear Reviewer, thank you for your comments that helped us to improve our manuscript.

The authors should add some more information about the study group, e.g. age, the severyty of acne (mild, moderate, severe), assessment of acne using global acne grading system (GAGS.

we added as requested these data in the text

Please, add inclusion and exclusion criteria

Inclusion and exclusion criteria have been added

Why the authors have assessed Fitzpatrick Wrinkle Assessment Scale (FWAS)? What was the aim of the study? Please explain more clearly in the text. All the patients had acne problemes or you treated patients with either acne or wrinkles?

We specified in the text that even if the primary aim of this study was to evaluate the role of carbon laser treatment for acne and pore reduction, a secondary aim was to evaluate the response of thin wrinkles; in fact stimulation of the dermis induced by laser-irradiated carbon shock waves can result in improvement of the wrinkles. To evaluate the efficacy of carbon peel laser, we used the FWAS to have a specific score for wrinkles evaluation.